# The Known and Unknown of Global Tick-Borne Viruses

**DOI:** 10.3390/v16121807

**Published:** 2024-11-21

**Authors:** Abulimiti Moming, Yuan Bai, Jun Wang, Yanfang Zhang, Shuang Tang, Zhaojun Fan, Fei Deng, Shu Shen

**Affiliations:** 1Key Laboratory of Virology and Biosafety and National Virus Resource Center, Wuhan Institute of Virology, Chinese Academy of Sciences, Wuhan 430071, China; abulimiti@wh.iov.cn (A.M.); by@wh.iov.cn (Y.B.); wangjun@wh.iov.cn (J.W.); zhangyf@wh.iov.cn (Y.Z.); ts@wh.iov.cn (S.T.); roy@wh.iov.cn (Z.F.); 2Center for Disease Control and Prevention of Xinjiang Uygur Autonomous Region, Urumqi 830002, China; 3Xinjiang Key Laboratory of Vector-Borne Infectious Diseases, Urumqi 830002, China; 4Hubei Jiangxia Laboratory, Wuhan 430200, China

**Keywords:** tick-borne viruses (TBVs), history of discovering TBVs, tick vectors, spillover, diseases caused by TBVs

## Abstract

Ticks are crucial vectors for various pathogens associated with human and animal diseases, including viruses. Nevertheless, significant knowledge gaps prevail in our understanding of tick-borne viruses (TBVs). We here examined existing studies on TBVs, uncovering 870 documented virus species across 28 orders, 55 families, and 66 genera. The discovery history, vector ticks, and hosts of TBVs, as well as the clinical characteristics of TBV-induced diseases, are summarized. In total, 176 tick species from nine tick genera were confirmed as vectors for TBVs. Overall, 105 TBVs were associated with infection or exposure to humans and animals. Of them, at least 40 were identified to cause human or animal diseases. This review addresses the current challenges associated with TBV research, including the lack of knowledge about the identification of novel and emerging TBVs, the spillover potentials from ticks to hosts, and the pathogenicity and infection mechanisms of TBVs. It is expected to provide crucial insights and references for future studies in this field, while specifically focusing on expanding surveys, improving TBV identification and isolation, and enhancing the understanding of TBV–vector–host interactions. All of these findings will facilitate the preparation for preventing and treating diseases caused by emerging and novel TBVs.

## 1. Introduction

With the rapid emergence and spread of previously unknown or neglected viruses, the vulnerability of global populations to the possible health and economic consequences of vector-borne viruses, such as Zika virus, dengue virus, yellow fever virus, and Crimean–Congo hemorrhagic fever virus (CCHFV), has become evident [1,2,3,4]. A lag in the development of effective vaccines or treatments was commonly observed after identifying these novel viruses, which require time for researchers and clinicians to overcome the viral diseases. Moreover, studies proactively investigating unknown or neglected viruses are scarce, which hinders the assessment of the potential transmission risks of those viruses, thereby impeding the implementation of effective measures for controlling virus outbreaks and leaving us only marginally prepared for the next epidemic [5]. Globally, it is estimated that approximately 1.67 million undiscovered viruses exist in animal reservoirs, with half of them capable of causing zoonotic diseases in humans and animals [6]. However, as our understanding of the prevalence, diversity, and ecology of viruses, as well as the driving factors underlying virus dissemination and evolution, is inadequate, our capacity to mitigate viral disease emergence is significantly limited. This underscores the need to discover and study effective control measures urgently to prevent future epidemics. Such measures include the discovery and characterization of novel viruses and their reservoirs, and an assessment of their transmission potentials and substantial risks. The viromes and the characteristics of viruses that can transmit from reservoirs, such as bats, birds, and rodents [7,8], or vectors, such as mosquitoes and ticks [9], to their hosts, remain insufficiently elucidated.

Ticks are a significant vector for transmitting viral pathogens associated with human and animal diseases and, thus, play a pivotal role in virus dissemination [10,11]. Tick-borne viruses (TBVs) are a group of viruses carried and transmitted by ticks. TBVs have been identified as pathogens infecting humans and/or animals for over a century [12]. Some of the most significantly pathogenic TBVs, including CCHFV, tick-borne encephalitis virus (TBEV), severe fever with thrombocytopenia syndrome virus (SFTSV), Nairobi sheep disease virus (NSDV), and African swine fever virus (ASFV), have exerted notable effects on public health and the economy [13,14,15,16,17]. As ticks transmit viral pathogens by taking blood meals from hosts, the increasing likelihood of ticks interacting with animals and humans, which is influenced by extensive geographical distribution and climate or ecologic changes, has the potential to escalate the risk of emerging diseases posed by rising TBV transmission [18,19].

Recently, the advent of advanced sequencing technology has notably increased TBV discovery. The metavirome analysis of diverse tick species from different locations unveiled a remarkable number of RNA viruses, which may have redefined the viral sphere [20,21]. However, this likely represents only a fraction of the total TBV community. An expansion of monitoring activities can facilitate the detection of additional TBVs, thereby underscoring the risk associated with human or animal exposure to these viruses. Additionally, the number of novel TBVs linked to human diseases is increasing, including the Alongshan virus (ALSV) [22], Yezo virus (YEZV) [23], Songling virus (SGLV) [24], Tacheng tick virus-1 (TcTV-1) [25], and Tacheng tick virus-2 (TcTV-2) [26], and this has become a matter of public concern. The pervasive prevalence of TBVs is a substantial public health challenge, necessitating the implementation of multifaceted approaches to address this issue and augment public awareness and preventive measures.

Although studies have investigated TBVs, they have not provided a comprehensive overview of their current global landscape. Consequently, a detailed understanding of global TBVs is essential for improving our comprehension of the pathogens, thereby facilitating the development of effective control strategies and mitigating TBV spread. This review aims to build a comprehensive understanding of global TBVs through a detailed analysis of their classification characteristics, discovery history, clinical manifestations of TBV-induced diseases, and public health implications.

## 2. Taxonomy of Currently Known TBVs

An inclusive literature search was conducted on the discovery and prevalence of TBVs and their potential association with human and/or animal diseases by referring to Web of Science (https://www.webofscience.com/, accessed on 30 December 2023) and PubMed (https://pubmed.ncbi.nlm.nih.gov/, accessed on 30 December 2023). A total of 595 articles related to ticks and TBVs were reviewed (Appendix A). In these studies, 870 virus species were documented, which were categorized across 3 kingdoms, 7 phyla, 15 classes, 28 orders, 55 families, and 66 genera (Figure 1 and Appendix A). *Hareavirales* and *Ourlivirales* are orders with the greatest number of TBVs, each containing 172 and 132 TBVs, respectively. Subsequently, *Monegavirales*, *Renovirales*, *Durnavirales*, *Wolframvirales*, *Ghabrivirales*, *Articuvirales*, *Amarillovivirales*, *Picornavirales*, *Cryptovirales*, and *Jingchuvirales* also contain a marked number of TBVs, ranging from 23 to 69 species. Additionally, *Toliviverales*, *Martellivirales*, *Tymovirales, Nodamuvirales*, *Patatavirales*, *Stellavirales*, *Nidovirales*, *Piccovirales*, *Sobelivirales*, and *Hepelivirales* contain TBV representatives, but in smaller numbers of <20 species. In contrast to the other orders, *Asfuvirales*, *Chitovirales*, *Cirlivirales*, and *Cryptovirales*, each contain a single TBV. Moreover, 97 TBVs were unassigned to any specific classification, possibly because of their diverse nature relative to other known viruses. These findings indicate the extensive and heterogeneous characteristics of the TBV community.

Most known TBVs are RNA viruses, which account for approximately 99% of the total (Appendix A). These RNA viruses include 152 double-stranded (ds) RNA viruses, 339 single-stranded positive-sense (ssRNA+) viruses, and 276 single-stranded negative-sense (ssRNA−) viruses, distributed across 26 orders and 53 families (Appendix A). The 152 dsRNA TBVs primarily belong to the orders *Reovirales*, *Durnavirales*, and *Ghabrivirales*. The 339 ssRNA+ viruses have been identified in numerous varying orders, including *Amarillovirales*, *Martellivirales*, *Nidovirales*, *Nodamuvirales*, *Picornavirales*, *Tolivirales*, *Tymovirales*, *Cryppavirales*, *Hepelivirales*, *Ourlivirales*, *Patatavirales*, *Sobelivirales*, *Stellavirales*, and *Wolframvirales.* The 276 ssRNA− TBVs primarily belong to the orders *Articulavirales*, *Hareavirales*, *Jingchuvirales*, and *Mononegavirales.* In addition to the classified TBVs, 95 TBVs remain unclassified as specific information regarding their RNA types is lacking (Appendix A). Of the total 870 TBVs documented, only two viruses are dsDNA viruses, as follows: ASFV [27], a member of the *Asfarviridae* family, and the Lumpy skin disease virus (LSDV) [28], a member of the *Poxviridae* family. Six ssDNA TBVs have also been identified, namely, Avian-like circovirus [29], from the family *Circoviridae*; Nayun tick torquevirus [30] and tick-associated torque teno virus [31], from the family *Anelloviridae*; and Roe deer copiparvovirus [32], Lone star tick densovirus [29], and Bovine hokovirus [33], all belonging to the family *Parvoviridae*. Most RNA and DNA TBVs have genomic lengths of 2000–30,000 nucleotides. Different from the others, the two dsDNA viruses, ASFV and LSDV, have markedly greater genomic lengths of 150,000–180,000 nucleotides (Appendix A).

## 3. Over a One-Hundred-Year History of Discovering TBVs

The Nairobi sheep disease virus (NSDV), described in Kenya in 1912, is the first reported TBV. This virus has caused viral disease in sheep with a high mortality rate [12]. However, the discovery of NSDV drew no significant attention to TBVs until the mid-20th century. Before the 1950s, only a few TBVs were identified, such as ASFV (1921), LSDV (1929), Louping ill virus (LIV, 1931), African horse sickness virus (AHSV, 1935), TBEV (1937), CCHFV (1944), and Omsk hemorrhagic fever virus (OHFV, 1947), which were discovered only because animals/humans were becoming ill, and not active discovery of TBVs supported by funding (Figure 2A and Appendix A). This slow progress in the discovery of novel TBVs could be attributed to the limited collection of tick or patient samples suitable for virus identification and isolation by using very limited methodology. Moreover, the progress was subsequently disrupted by World War II (1931–1945). A major upsurge in the study of TBVs was noted from the 1950s to the 1970s, which was supported by a comprehensive Rockefeller Foundation (RF) virus program [34,35]. This program facilitated the discovery and characterization of new TBVs in Africa and India, including the Bhanja virus (BHAV, 1954), Kyasanur forest disease virus (KFDV, 1957), Quaranfil virus (QRFV, 1960), and Dhori virus (DHOV, 1961) [34]. During this period, the morphology, serologic category, and virulence of these TBVs in animals were characterized using methods such as serological testing, electron microscopy, and animal infection assays (Appendix A). The use of appropriate methodologies, sustained commitment, and financial support have been instrumental in facilitating TBV identification. Since the 1980s, technological advances such as Sanger sequencing and polymerase chain reaction have facilitated TBV identification by allowing the quick discovery of TBV sequences from any samples, unlike methods that require extensive efforts to obtain virus isolates. The metagenomic sequencing technology [20,21], which allows for the identification of numerous viral sequences from various biomaterials, including ticks, has further revolutionized TBV discovery, thereby leading to an upwell of TBV sequence data. To date, 683 TBVs (79%) of the total have been identified through metagenomic sequencing, predominantly including TBVs belonging to *Ourlivirales* (131 species), *Hareavirales* (80 species), *Monongavirales* (58 species), and *Durnavirales* (49 species). The remaining 187 TBVs were identified through isolation by using tick or patient samples and through molecular and/or serological tests. According to the taxonomy, these TBVs belonging to *Hareavirales* had the highest diversity as they comprised 87 viral species, followed by *Reovirales* with 47 viral species (Figure 2B). This indicates epidemic exposure in TBVs belonging to *Hareavirales* and *Reovirales*, which have a greater potential for transmission. TBVs belonging to *Cirlivirales, Cryptovirales, Durnavirales, Ghabrivirales, Hepelivirales, Jingchuvirales, Martellivirales, Nidovirales, Nodamuvirales, Ourlivirales, Patatavirales, Piccovvirales, Sobelivirales, Stellavvirales, Tolivivirales*, *Tymovirales*, and *Durnavirales* were identified only through sequencing and in the absence of serological evidence or isolates. Thus, the epidemiological risk assessment of these viruses may be inadequate, or some viruses may not be capable of cross-species transmission and, thus, belong to endogenous tick viruses. As virus isolates are lacking, the morphology and infectious properties of most TBVs identified only through sequencing remain unclear. These concerns can be addressed by improving existing methods or developing new methods for virus isolation.

## 4. Ticks to Carry Viruses

Ticks (Acari: *Ixodidae*) are blood-sucking ectoparasites that have existed since the Mid-late Cretaceous era and have undergone substantial evolutionary adaptations [36]. Ticks can parasitize various vertebrates, even extinct reptiles such as dinosaurs [37]. They can be found in diverse habitats worldwide, even in extremely cold environments such as Antarctica [38,39]. According to species diversity, *Ixodoidea* is divided into three families based on their morphological characteristics, as follows: the well-known *Ixodidae* and *Argasidae*, and the rarely noticed *Nuttallielidae* [40]. To the best of our knowledge, 911 tick species are present in the world [41,42]. The *Ixodidae* family is the most diverse and widespread and comprises 709 tick species, which is considerably higher than those present in the *Argasidae* and *Nuttallielidae* families. *Argasidae* contains 201 species, and *Nuttallielidae* has a single species, *Nuttelliella Namaqua*, observed only in Europe [43].

The tick life cycle consists of the following four stages: egg, larva, nymph, and adult. For the complete transformation from larva to adult, ticks require blood meals from animal hosts [44]. Because ticks can carry numerous viruses throughout their life cycle, they act as long-term viral vectors. Moreover, ticks may be able to introduce TBVs into various animal hosts, including mammals, reptiles, amphibians, and birds, during a blood meal [45]. In total, 176 tick species from nine genera (*Amblyomma*, *Antricola*, *Argas*, *Dermacentor*, *Haemaphysalis*, *Hyalomma*, *Ixodes*, *Ornithodoros*, and *Rhipicephalus*) have been confirmed as viral vectors (Figure 3 and Appendix A). The genus *Ixodes* is a large community comprising 246 identified tick species, markedly more than the species identified in other genera. Of them, 32 *Ixodes* species (13% of the total) were virus vectors. Although the genera *Hyalomma* and *Dermacentor* contain 27 and 34 identified tick species, respectively, representing a smaller species population than the genus *Ixodes*, 20 and 18 of their species can carry viruses, representing 75% and 53% of the total species in the two genera. This indicates that *Hyalomma* and *Dermacentor* ticks play a significantly more crucial role in carrying viruses than the other tick genera. This also suggests that the greater risks of virus spread in areas where these ticks are prevalent. A few tick species in other genera, including *Antricola*, *Argas*, *Ornithodoros*, *Amblyomma*, *Haemaphysalis*, and *Rhipicephalus*, were also found to carry viruses. They compose <31% of the total species of each genus, considerably less than those present in *Hyalomma* and *Dermacentor*. *Ixodidae* ticks generally harbor a greater number of TBVs than *Argasidae* ticks. Nevertheless, relevant reports or studies are missing for more than 700 tick species, rendering it uncertain whether they can carry and transmit viruses. Ongoing and sustained surveys for viruses in various tick species may affect the rates that indicate the significance of each tick genus as a TBV vector.

The geographic distribution of specific tick species may influence TBV dissemination, as they may serve as vectors for only certain TBVs. For example, viruses belonging to *Chitovirales* have been identified in *Amblyomma* and *Rhipicephalus* ticks, and *Asfuvirales* and *Cirlivirales* have only been found in *Ornithodoros* and *Ixodes*, respectively. As these TBVs have only been found in a limited range of tick species, they may have a relatively narrow vector specificity. By contrast, viruses from *Articulavirales*, *Hareavirales*, and *Mononegavirales* exhibit broad specificity and are found in tick species from all nine tick genera (Appendix A). The correlation between tick species and specific viruses, at the family, genus, or species level, is crucial for the accurate inspection and control of TBV spread and the assessment of the potential risk of spillover. Nevertheless, despite this correlation being significant, it has remained vague and imprecise.

## 5. TBV Spillover to Hosts

Among the 870 known TBVs, 105 (12%) viruses belonging to 11 families, namely, *Asfarviridae*, *Flaviviridae*, *Nairoviridae*, *Orthomyxoviridae*, *Parvoviridae*, *Peribunyaviridae*, *Phenuiviridae*, *Poxviridae*, *Sedoreoviridae*, *Rhabdoviridae*, and *Tobaniviridae*, were identified as viral pathogens that were associated with human and/or animal infections (Figure 4). Based on reports providing results obtained using virus isolates, as well as evidence from serologic or nucleic acid tests, 59 TBVs can cause infection in humans, 63 TBVs in domestic animals, 51 TBVs in birds, 25 TBVs in rodents, and 16 TBVs in wild animals (Appendix A). TBVs from *Nairoviridae*, *Flaviviridae*, *Orthomyxoviridae*, *Peribunyaviridae*, *Phenuiviridae*, *Sedoreoviridae*, and *Rhabdoviridae* were found to be associated with human infections, and TBVs from *Asfarviridae*, *Poxviridae*, *Parvoviridae*, and *Tobaniviridae* were exclusively reported to infect domestic animals.

Whether the other 765 TBVs can infect or spread from ticks to hosts is currently unclear. Nevertheless, some of them may have the potential to infect or spread. This can be demonstrated by increasing the number of surveys in more wide regions, identifying cases of infection, and obtaining virus isolates. Unlike TBVs identified following the detection of diseases in the last century, TBVs such as TcTV-1 and TcTV-2 were first identified through sequencing and subsequently found to be associated with human diseases, because these viruses were isolated from samples of febrile patients [25,26]. In addition to viruses with the potential to spill over from ticks to hosts, which are yet to be identified, endogenous viruses are persistently carried by ticks. These endogenous viruses may not be able to spread to animals and/or humans. St Croix River virus (SCRV) [46,47], an *orbivirus* (*Sedoreoviridae*), has been exclusively found in tick cell lines, such as IDE2 and IDE8 derived from *Ixodes scapularis*, and RA243 and RA257 from *Rhipicephalus appendiculatus* [48]. SCRV cannot infect any mosquito- or mammal-derived non-tick cells [47]. Nevertheless, the role of these endogenous viruses in ticks remains uncertain. The impact of these viruses on the tick life cycle and the potential role of ticks in the transmission of viral pathogens to hosts must be elucidated. Moreover, whether these endogenous viruses will eventually evolve the capacity to be transmitted from ticks to animals over their long-term evolution needs to be determined.

## 6. TBVs Associated with Human and Animal Diseases

Some TBVs can cause diseases and, therefore, always garner attention immediately after the diseases have been noted. At least 40 TBVs, primarily belonging to *Flaviviridae, Nairoviridae*, and *Phenuiviridae*, cause diseases in both humans and animals (Figure 5). Patients with TBV-induced diseases typically exhibit febrile symptoms (fever, headache, fatigue, chills, and dizziness), which are mostly accompanied by neurologic manifestations (encephalitis, mental confusion, depression, malaise, and meningitis); gastrointestinal symptoms (anorexia, vomiting, nausea, abdominal pain, abdominal distention, diarrhea, and melena); hemorrhagic signs (hemorrhage, hypotension, mucous hemorrhage, hematochezia, hematemesis, and thrombocytopenia); respiratory disorders (cough, and respiratory disorders); and urinary symptoms (hematuria and oliguria). Other manifestations, such as myalgia, arthralgia, rash or petechiae, and conjunctival hyperemia, are occasionally presented by the patients. TBVs belonging to the *Flaviviridae* family that cause diseases are predominantly transmitted by *Ixodes* ticks. On the other hand, pathogenic TBVs belonging to the *Nairoviridae* family are primarily transmitted by *Haemophilus* ticks (Figure 5 and Appendix A).

The number of TBVs that can infect animals and/or humans and cause diseases in them is far lower than the total number of known TBVs. This discrepancy leaves considerable uncertainty and poses major challenges to public health. Not all TBVs seem to have the ability to induce infection-related disorders, which may result in a fatal outcome. Of the 40 human- and/or animal disease-causing TBVs, 13 TBVs can cause fatal diseases. These include Alkhumra hemorrhagic fever virus, Kyasanur forest disease virus (KFDV), LIV, OHFV, West Nile virus, Powassan virus, and TBEV belonging to *Flaviviridae*; ASFV belonging to *Asfarviridae*; CCHFV and NSDV belonging to *Nairoviridae*; Rift Valley fever virus and SFTSV belonging to *Phenuiviridae*; and LSDV belonging to *Poxviridae*. Of these, CCHFV has been reported in tick species belonging to eight genera from the *Ixodoidea* family. These TBVs represent a considerably more diverse range of tick vectors that carry this virus than the other TBVs (Figure 5 and Appendix A).

## 7. Challenges and Opportunities in Research of TBVs

Growing evidence about pathogenic TBVs has offered invaluable insights into their transmission, pathogenesis, and impact on global health. Nevertheless, these data remain insufficient to allow us to prepare to efficiently tackle the potential global impact of emerging or unknown TBVs. Unknown aspects of TBVs remain, and these will continue to challenge different TBV research fields for an extended period (Figure 6).

First, although numerous TBVs were identified, which markedly increased the total number of identified viral species, most of these discoveries were confined to geographically limited and/or specific regions. These findings typically focused on dominant tick species or were derived from non-systematic sampling surveys. Ticks are distributed globally, but numerous regions remain to be investigated for the presence of ticks. In these regions, the viromes of ticks remain unknown or have been explored only to a limited extent, possibly because specific tick samples and/or adequate financial and technical support to conduct surveys is unavailable. The most recently discovered TBVs are mainly reported from North American, European, and Asian countries, including the United States [49], China [20,21], and Turkey [50]. These reports are typically based on metagenomic sequencing results. By contrast, TBVs in other regions, particularly from Africa and South America, tend to be reported after the infected cases are detected (Appendix A), along with very few preemptive investigations of the tick viromes present in those regions [51,52]. China has recently contributed remarkably to exploring the great TBV community by conducting comprehensive surveys on different tick species across various habitats. Consequently, over 500 new TBVs were identified through metagenomic sequencing [20]. Expanding the scope of these surveys is imperative to achieve a comprehensive understanding of global TBVs, including their diversity, distribution, and associations with tick species.

Second, whether all known tick species can serve as vectors carrying and transmitting viruses remain unelucidated. To date, 176 tick species are known to serve as carriers of viruses, representing <20% of the total 911 species recorded (Figure 3). This implies that >80% of tick species remain to be investigated. Moreover, epidemiological data regarding known TBVs are insufficient to generate a model for evaluating TBVs’ potential to spill over from ticks to different hosts or for fully understanding their transmission pattern. A major proportion of ticks is known to move with their hosts, such as migratory birds [53,54], wildlife [55], and livestock [56]. Among these hosts, birds are possibly a crucial player in the dynamics of spreading ticks and tick-borne diseases. Birds are hosts for various tick species and aid in transporting ticks across short and long distances, even spanning between continents [54]. Furthermore, the increasing impacts of environmental and climate changes on the dynamics between migratory animal hosts, ticks, and TBVs remain to be elucidated. The contribution of this tick movement to TBV spread between different regions and countries remain uncertain, despite such migration of a few TBVs, such as CCHFV [13,57], SFTSV [58], and TBEV [59,60], being characterized based on virus phylogeny. A more detailed understanding of the association between a specific TBV or categorized TBVs and tick species, along with a meticulous examination of the vector’s living habits and habitats, is required. This would improve our knowledge of TBV transmission patterns and facilitate the implementation of effective risk control measures against pathogenic TBVs.

Third, knowledge about viral pathogenicity and infection mechanisms is limited to TBVs, which cause severe human and animal diseases. These diseases exert a deep impact, and their complete understanding is lacking. Mechanisms underlying diseases caused by TBVs with a close phylogenetic relationship or belonging to the same species of different genotypes/subtypes may be similar. However, they could still demonstrate different virulence-related specific mechanisms among them, resulting in mild or severe infection [61,62]. For instance, SFTSV and Heartland virus (HRTV) are closely related genetically and are both classified under the *Bandavirus* genus (family: *Phenuiviridae*) [63,64]. Notwithstanding these similarities, discernible differences are noted in their infection mechanisms, mostly observed in the non-structural (NS) protein. In SFTSV, NS blocks signaling cascades by sequestering of host proteins into viral inclusion bodies [65]. By contrast, NS in HRTV does not form inclusion bodies, but it interacts with host proteins and restricts the induction of type-I interferons [66]. Moreover, according to experimental studies, in immunocompetent C57BL/6 mice inoculated with SFTSV, the virus replicates to a detectable level and induces lesions in the spleens [67]. By contrast, experimentally inoculated HRTV in the same mouse model resulted in no lesions or detectable viremia [68]. The Langat virus (LGTV) is closely related to TBEV and exhibits 82%–88% amino acid homology [69]. TBEV can cause severe human infections, characterized by fever, encephalitis, and even death [70]. However, LGTV is non-pathogenic or less pathogenic to humans, which renders it an attractive candidate for TBEV vaccine development [62]. Consequently, to understand various TBVs, their common features must be first identified, followed by ascertaining the unique specifics of each TBV so as to develop more efficacious intervention and control strategies (Figure 6).

Opportunities are often accompanied by challenges. First, for a deeper understanding of the global TBV community, continuous surveys of TBVs must be conducted in a wider range of tick species and expanded geographic regions (Figure 6). A report on the viromes of ticks collected from Antarctic penguins added to our existing knowledge of ticks and TBV communities distributed from continents to the polar region such as Antarctica [71]. Consequently, regions that have not been explored can be investigated, or tick communities can be monitored in regions where the TBV dissemination characteristics remain ambiguous. Such surveys reveal the abundance, diversity, and prevalence of TBV communities and elucidate their association with different tick species or probably diverse ecological environments. Accumulating vast amounts of data on TBVs and mapping their distribution and prevalence across various tick populations worldwide have led to a comprehensive understanding of global tick viromes. These data provide a critical basis for further evaluating spillover risks, predicting transmission events, and preventing or controlling TBV-induced outbreaks.

Second, major advances in sequencing techniques have markedly augmented the capacity to discover and identify novel viruses (Figure 6). Applying these advances to TBV identification has provided insights into their genomic structure, evolution, and diversity. Bioinformatic analyses based on TBV genome sequences identified mutation, recombination, and re-assortment patterns, thereby resulting in the emergence of new strains or novel viral species. TBV isolates are a significant prerequisite for systematic research on infection mechanisms and pathogenesis. The techniques most commonly used for isolating viruses from ticks are as follows: (1) Tick homogenates are incubated with mammalian cells, and the mixture of cells and supernatants are subcultured (blind passage) until cytopathic effects (CPEs) are observed. Alternatively, immunofluorescence assays are conducted to identify virus isolation in the absence of CPEs [72,73]. (2) Tick homogenates are inoculated into suckling mice. Then, the brains of diseased mice are collected and subjected to metagenomic analyses to identify the virus at a high abundance, which suggests successful infection and replication of the tick-derived virus. Further, TBV-positive mouse brain homogenates are incubated with human or mammalian cells to obtain virus isolates [74]. The etiological properties and pathogenesis of the TBV isolates are then characterized using sensitive cell lines and animal models. This will enable the investigation of TBV infectivity, replication cycle, potential host range, and virulence.

Third, from a medical perspective, knowing TBV characteristics and life cycle activities at the cellular level in detail is crucial, as is identifying potential targets for antiviral drug development. Investigating host cell responses triggered by virus infection, such as interferon production, inflammatory reactions, cell apoptosis, and alterations in cell survival pathways, is also critical. These findings collectively portray the intricacies of virus–host interactions, clarifying the intracellular battle between viruses and hosts. Based on these observations, molecular mechanisms underlying TBV pathogenesis can be delineated and key factors influencing viral and host outcomes can be identified (Figure 6). Further studies will necessitate experimentation at the animal level. An animal infection model needs to be established to mimic natural TBV infections. These models will allow systematic characterization of the tissue tropism of TBV infection and the innate and adaptive immune responses to specific TBVs. This information can be used to advance TBV prevention strategies; formulate and evaluate drugs that can impede the virus life cycle or bolster the host’s antiviral capacity; and create vaccines that can stimulate a protective immune response in hosts.

To successfully circulate within natural reservoirs, TBVs must sustain within the tick vector for an extended period. Understanding the impact of prolonged TBV infection on ticks, and elucidating mechanisms underlying the carriage and transmission of multiple TBVs by ticks are markedly important (Figure 6). Therefore, to elucidate TBV–vector interactions, life cycle events of TBVs and molecular responses to them in tick bodies must be investigated using specific biomaterials, including tick cell lines and feeding tick-based infection models. However, very few studies have conducted TBV infection experiments using tick cell lines and tick-feeding models. In a study, CCHFV was replicated in seven different tick cell lines to varying degrees. Accordingly, the study proposed a possible species–specific limitation of CCHFV infection in different ticks [75]. Another study reported that TBEV can replicate and establish persistent infections in both ticks and tick cell lines. Continuous TBEV infection is beneficial for selecting virus variants with low neuroinvasiveness in laboratory mice [76]. These findings highlight the complexity of virus–vector interactions and the potential implications for viral adaptation and transmission in natural reservoirs. TBV infection alters the cellular response of tick cells. TBEV infection in the *I. scapularis*-derived tick cell line IDE8 and the *I. ricinus*-derived cell line IRE/CTVM19 induced significant expression of RNA transcripts and proteins involved in innate immune and cell stress responses [77]. In another study, the researchers investigated the genome plasticity of CCHFV by examining the virus in a model of tick–mouse transmission cycle. *Hyalomma marginatum* ticks were fed on CCHFV-infected mice, and were allowed to complete their feeding and molting, thus entering the next life cycle. A significant number of non-synonymous mutations in CCHFV genes, especially the viral glycoprotein genes, were detected within ticks after a single transstadial transmission, which were not observed in CCHFV recovered from mice [78]. It suggests that the genetic variations in TBVs occurred in ticks may be related to its life cycle, which may have an impact on the adaptability and transmission of TBVs.

## 8. The Future Perspective for Overcoming TBV Infection

Future research prospects should prioritize exploring potential treatments, prevention, and monitoring strategies, vaccine development, drug discovery, and antibody production for TBVs. Comprehensive studies on TBVs can offer invaluable insights, thereby allowing us to better address this major viral community, by ensuring global health and welfare. One promising research avenue is the exploration and development of TBV-targeting antiviral drugs. Understanding the critical aspects of viruses’ life cycle and their interaction with host molecules can provide potential drug targets. Developing effective vaccines that stimulate protective immune responses in the host is another valuable contribution to TBV management. Of note, several vaccines are already available for certain TBVs [79]. However, more broad-spectrum vaccines covering a wider array of TBVs would considerably improve our preparatory and reactive measures in controlling TBV effects globally.

Similarly, more effective and comprehensive methodologies must be developed for monitoring TBV distribution and prevalence worldwide. This can be achieved through advancement in global surveillance systems, including the development of portable diagnostic kits, the establishment of a global TBV database, and the encouragement of international collaboration to harmonize surveillance efforts. Meanwhile, developing prevention strategies remains a crucial component for controlling TBVs. Such guidelines, which include measures for effective tick control strategies, public health education on avoiding tick exposure, use of protective clothing and insect repellents, and regular tick inspection and removal of ticks, can help minimize tick exposure and, thus, reduce the risk of infection.

Finally, measures should be taken to further augment our understanding of TBV biology and TBV interaction with both hosts and vectors. This knowledge can provide a more precise scientific basis for developing effective vaccines and antiviral drugs and offer insights into TBV transmission dynamics and the risks of TBV infection to global health. In conclusion, the TBV-posed global threat indicates that continued research and surveillance efforts are required. As our understanding of TBVs is expanding, we must extend our toolkit of techniques for their early detection, control, and treatment. This will facilitate more effective preparation for and prevention of future TBV outbreaks, as well as a reduction in the impacts of these outbreaks on human health and well-being. The challenge is considerable, but so too are the technological and intellectual resources that can be used to resolve this global issue.

## Figures and Tables

**Figure 1 viruses-16-01807-f001:**
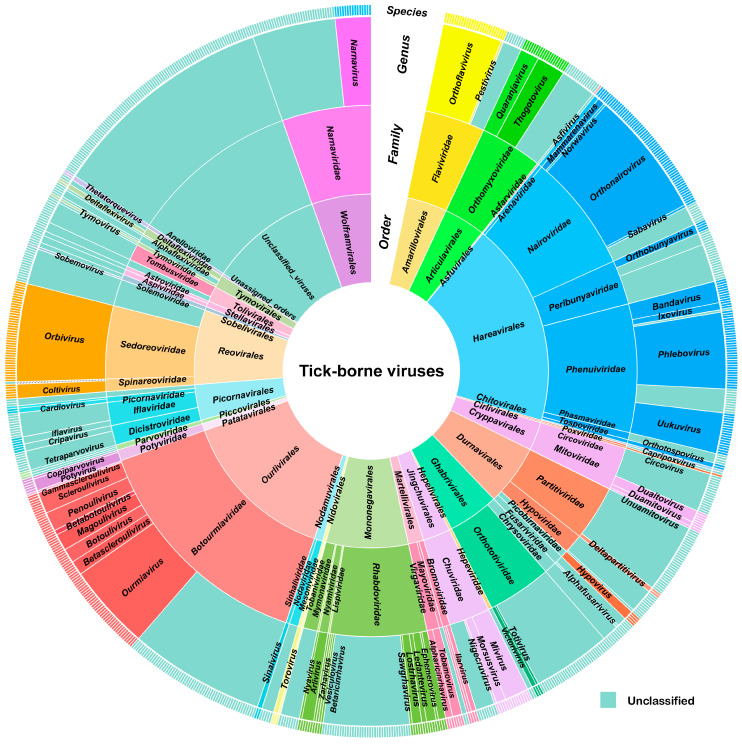
The taxonomy of globally reported TBVs. The size of each section is proportional to the percentage of species for TBVs at the order, family, and genus levels. Specifically, the species section represents one particular species of TBVs. The light aqua areas indicate unclassified TBVs within each category.

**Figure 2 viruses-16-01807-f002:**
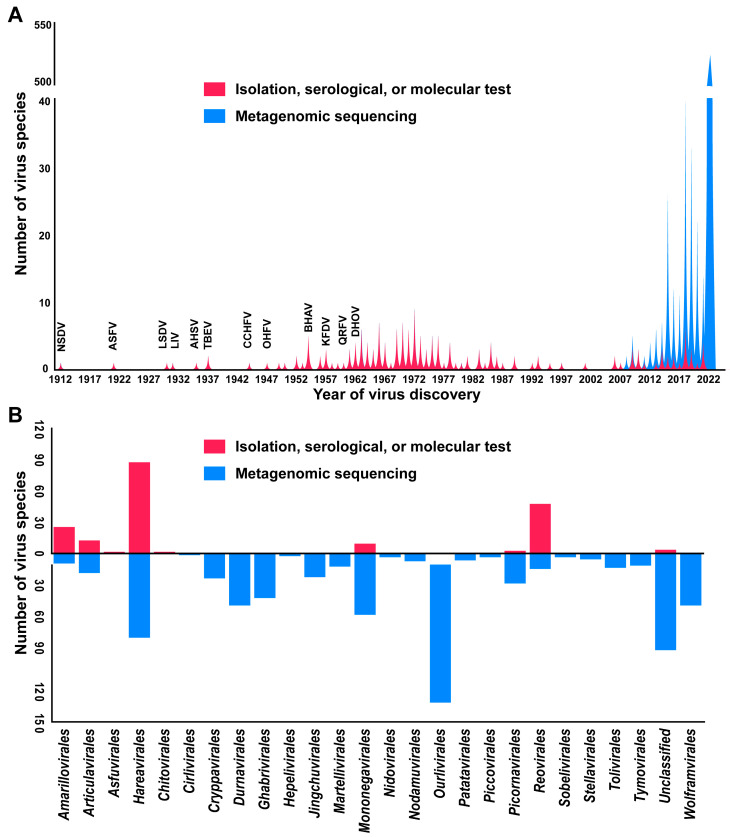
A timeline of the discovery of TBVs (**A**) and the categories of TBVs belonging to orders and distinguished according to the methodologies employed to discover them (**B**). The peaks and bars in red represent TBVs that were discovered by virus isolation, serological, or molecular tests; the peaks and bars in blue indicate TBVs that were discovered via metagenomic sequencing.

**Figure 3 viruses-16-01807-f003:**
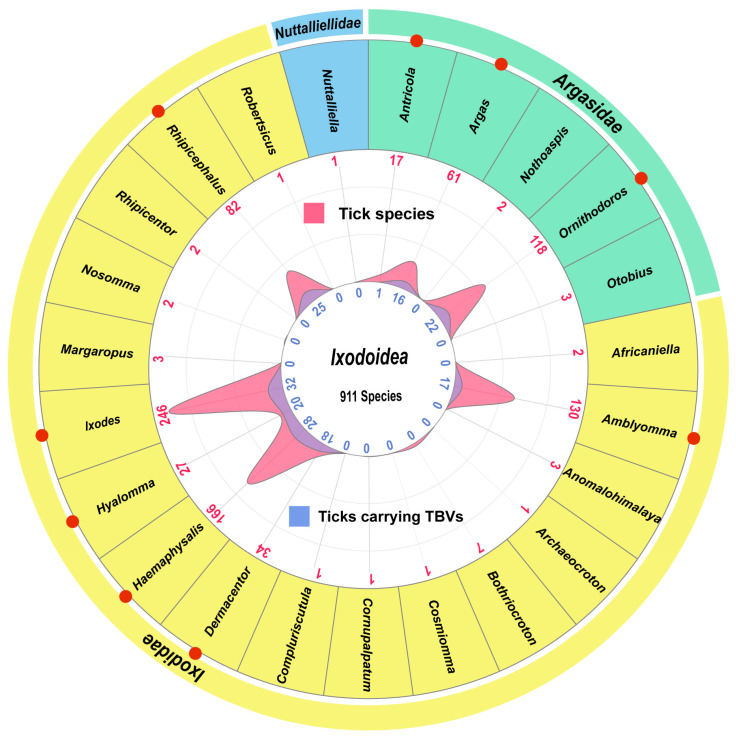
Ticks that are associated with TBVs. *Ixodoidea* includes 911 species of ticks belonging to 23 genera across the following three families: *Ixodidae*, *Argasidae*, and *Nuttallielidae*. The red dots represent tick genera that have been reported to carry TBVs, the red peaks and numbers represent the number of tick species in each genus, while the blue peaks and numbers represent the number of tick species in the corresponding genus that can carry TBVs.

**Figure 4 viruses-16-01807-f004:**
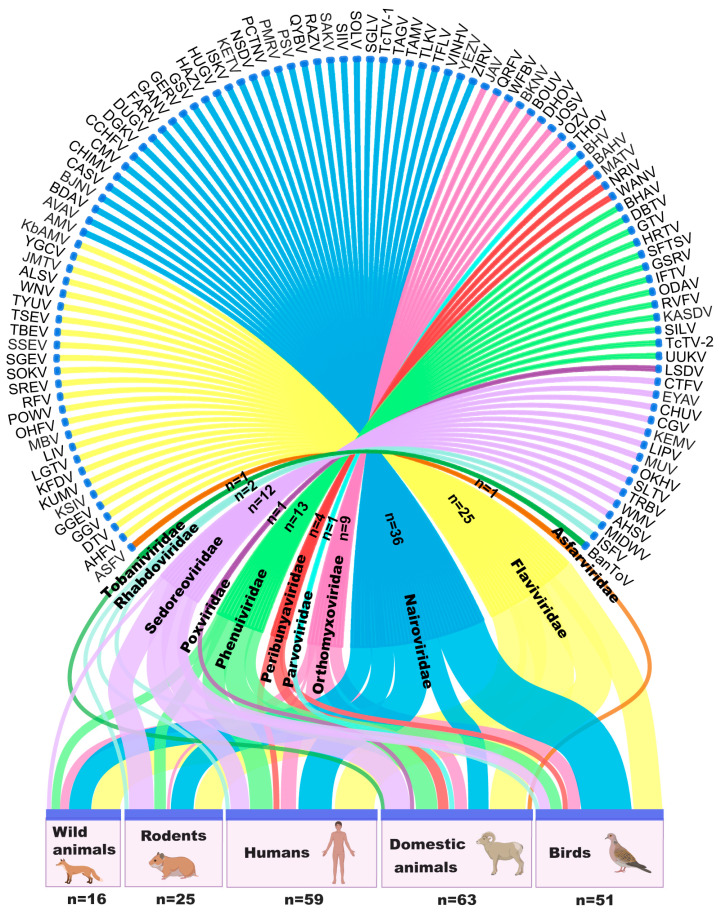
The TBVs that infect human and animal hosts. A total of 105 TBVs from 11 families associated with infections in humans and/or animals, including 1 TBV in *Asfarviridae*, 25 TBVs in *Flaviviridae*, 36 TBVs in *Nairoviridae*, 9 TBVs in *Orthomyxoviridae*, 1 TBV in *Parvoviridae*, 4 TBVs in *Peribunyaviridae*, 13 TBVs in *Phenuiviridae*, 1 TBV in *Poxviridae*, 12 TBVs in *Sedoreoviridae*, 2 TBVs in *Rhabdoviridae*, and 1 TBVs in *Tobaniviridae.* AHFV, Alkhumra hemorrhagic fever virus; AHSV, African horse sickness Virus; ALSV, Alongshan virus; AMV, Abu Mina virus; ASFV, African swine fever virus; AVAV, Avalon virus; BAHV, Bahig virus; BanToV, Bangali torovirus; BDAV, Bandia virus; BHAV, Bhanja virus; BHV, bovine hokovirus 2; BJNV, Beiji nairovirus; BKNV, Batken virus; BOUV, Bourbon virus; CASV, Caspiy virus; CCHFV, Crimean–Congo hemorrhagic fever virus; CGV, Chobar Gorge virus; CHIMV, Chim virus; CHUV, Chenuda virus; CMV, Clo Mor virus; CTFV, Colorado tick fever virus; DBTV, Dabieshan Tick Virus; DGKV, Dera Ghazi Khan virus; DHOV, Dhori virus; DTV, deer tick virus; DUGV, Dugbe virus; EYAV, Eyach virus; FARV, Farallon virus; GANV, Ganjam virus; GERV, Geran virus; GGEV, Greek goat encephalitis virus; GGV, Gadgets Gully virus; GSRV, Gissar virus; GSV, Great Saltee virus; GTV, Guertu virus; HAZV, Hazara virus; HRTV, Heartland virus; HUGV, Hughes virus; IFTV, Iftin tick virus; ISFV, Isfahan virus; ISKV, Issyk-Kul virus; JAV, Johnston Atoll virus; JMTV, Jingmen tick virus; JOSV, Jos virus; KASDV, Kaisodi virus; KbAMV, Kabuto mountain virus; KEMV, Kemerovo virus; KETV, Keterah virus; KFDV, Kyasanur forest disease virus; KSIV, Karshi virus; KUMV, Kumlinge virus; LGTV, Langat virus; LIPV, Lipovnik virus; LIV, louping ill virus; LSDV, lumpy skin disease virus; MATV, Matruh virus; MbATV, Mbalambala tick virus; MBV, Meaban virus; MIDWV, Midway virus; MUV, Muko virus; NRIV, Ngari virus; NSDV, Nairobi sheep disease virus; ODAV, Odaw virus; OHFV, Omsk hemorrhagic fever virus; OKHV, Okhotskiy virus; OZV, Oz virus; PMRV, Paramushir virus; POWV, Powassan virus; PSV, Punta salinas virus; QRFV, Quaranfil virus; QYBV, Qalyub virus; RAZDV, Razdan virus; RAZV, Raza virus; RFV, Royal Farm virus; RVFV, Rift Valley fever virus; SAKV, Sakhalin virus; SFTSV, severe fever with thrombocytopenia syndrome virus; SGEV, Spanish goat encephalitis virus; SGLV, Songling tick virus; SIIV, Sapphire II virus; SILV, Silverwater virus; SLTV, Seletar virus; SOKV, Sokoluk virus; SOLV, Soldado virus; SREV, Saumarez Reef virus; SSEV, Spanish sheep encephalitis virus; TAGV, Taggert virus; TAMV, Tamdy virus; TBEV, tick-borne encephalitis virus; TcTV-1, Tacheng tick virus 1; TcTV-2, Tacheng tick virus 2; TFLV, Tofla virus; THOV, Thogoto virus; TLKV, Tillamook virus; TRBV, Tribec virus; TSEV, Turkish sheep encephalitis virus; TYUV, Tyuleniy virus; UUKV, Uukuniemi virus; VINHV, Vinegar Hill virus; WANV, Wanowrie virus; WFBV, Wellfleet Bay virus; WMV, Wad Medani virus; WNV, West Nile virus; YEZV, Yezo virus; YGCV, Yamaguchi virus; ZIRV, Zirqa virus.

**Figure 5 viruses-16-01807-f005:**
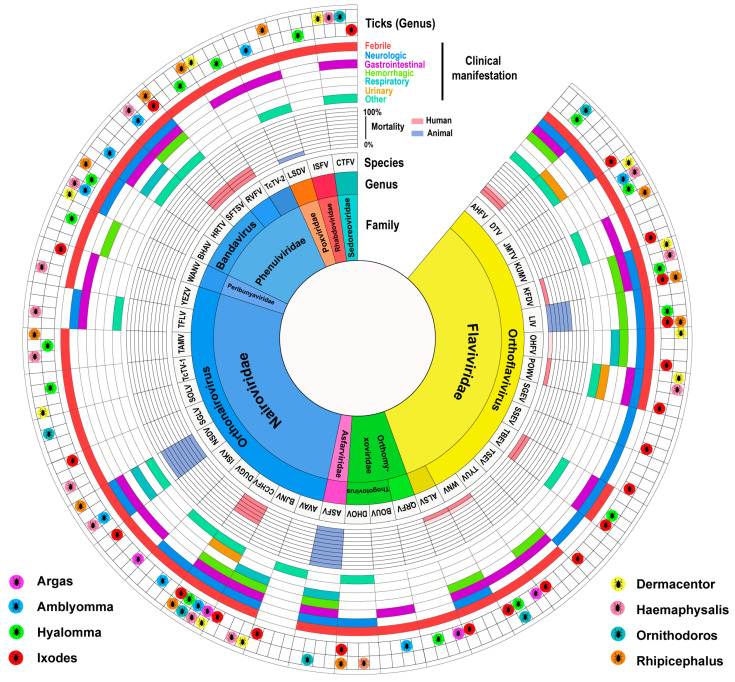
Pathogenic TBVs are associated with human or animal diseases. The circles represent, from inner to outer, the taxonomic classification at the family and genus levels of TBVs, the species of TBVs, mortality rate, clinical manifestations, and the vector ticks of TBVs.

**Figure 6 viruses-16-01807-f006:**
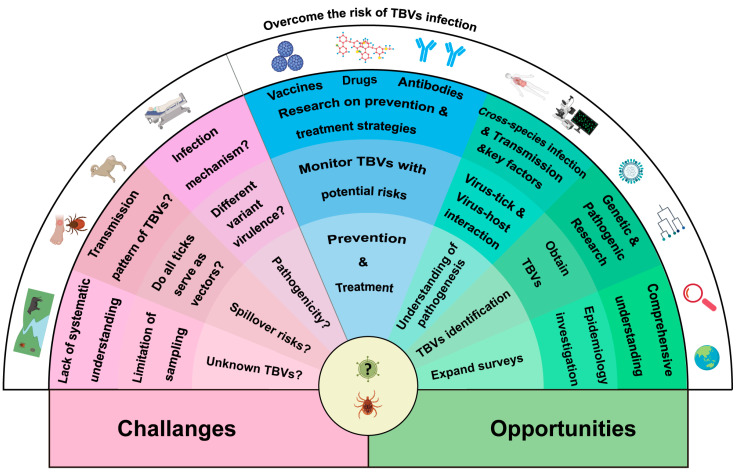
Challenges and opportunities in the research of known and unknown TBVs.

## Data Availability

All data presented and described in this manuscript are available in the database of the National Virus Resource Center (NVRC), Wuhan Institute of Virology, Chinese Academy of Sciences (identifier CSTR:16533.11.99.tick.host).

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
