# Peer review of "The Known and Unknown of Global Tick-Borne Viruses"

_viruses, 2024, doi:10.3390/v16121807_

Round 1

Reviewer 1 Report

Comments and Suggestions for Authors

The review manuscript by Moming et al. entitled “The known and unknown of global tick-borne viruses” summarized current situation of studies about TBVs identified so far. This review will undoubtedly provide important and useful clues for researchers of TBVs and other infectious diseases to proceed further studies, however, this reviewer raises some concerns as enumerated below:

1. Line 87, 150 and others: Buyavirales are reclassified as Bunyaviricetes including Elliovirales and Hareavirales by updated classification of ICTV. Authors should modify these descriptions throughout the text.

2. Line 87, 121 and others: Does supplementary file include only supplementary Figure 1 of network diagram of TBVs? This figure just shows correlations of TBVs and their estimated host species, but not indicate genomic length (line 121), detection methods (line 140), classification and tick vectors (line 310 and 326), cases and geographical information (line 346).

3. Figure 2: It seems to be confusing for readers to understand. Figure 2A and 2B should me clearly separated. The bottom line of years looks to indicate Figure 2B, too.

4. Figure 4: Some abbreviations are missing, e.g. TFLV.

Author Response

The review manuscript by Moming et al. entitled “The known and unknown of global tick-borne viruses” summarized current situation of studies about TBVs identified so far. This review will undoubtedly provide important and useful clues for researchers of TBVs and other infectious diseases to proceed further studies, however, this reviewer raises some concerns as enumerated below:

Comments 1. Line 87, 150 and others: Buyavirales are reclassified as Bunyaviricetes including Elliovirales and Hareavirales by updated classification of ICTV. Authors should modify these descriptions throughout the text.

Response 1: Thank you for pointing this out. We have updated the classification of TBVs throughout the manuscript based on the latest ICTV catalog.

Comments 2. Line 87, 121 and others: Does supplementary file include only supplementary Figure 1 of network diagram of TBVs? This figure just shows correlations of TBVs and their estimated host species, but not indicate genomic length (line 121), detection methods (line 140), classification and tick vectors (line 310 and 326), cases and geographical information (line 346).

Response 2: Thank you for pointing this out. The supplementary materials for this manuscript consist of Supplementary Table 1 and Supplementary Figure 1. Supplementary Figure 1 presents a network diagram of TBVs, demonstrating the evidence of infection or exposure in human and animal hosts. In Supplementary Table 1, details are provided regarding the genomic length, detection methods, classification, tick vectors, cases, and geographic information of TBVs. To ensure clarity and avoid ambiguity, the supplementary file in the manuscript has been revised and renamed as Supplementary Table 1.  

Comments 3. Figure 2: It seems to be confusing for readers to understand. Figure 2A and 2B should me clearly separated. The bottom line of years looks to indicate Figure 2B, too.

Response 3: Thank you for good reminding.We agree with this comment. Therefore, we have separated Figure 2A and 2B in the revised manuscript.

Comments 4. Figure 4: Some abbreviations are missing, e.g. TFLV.

Response 4: Thank you for good reminding. We have included missing abbreviations in the revised manuscript such as “TFLV, Tofla virus” on line 276.

Reviewer 2 Report

Comments and Suggestions for Authors

Comments to the Authors

This manuscript is valuable and well written.  The authors present a comprehensive review on tick-borne viruses (TBV), with particular attention on the TBV taxonomy and their association with animal diseases.  The figures were particularly interesting and provide a nice overview of the current standing of TBVs from pathogenic and taxonomic perspectives.  This reviewer has no major comments.  Minor comments/suggestions are listed below.

Minor comments:  
Lines 28-48: The introductory paragraph seems tangential to the topic at hand.  Viruses such as Ebola and Middle East respiratory syndrome are transmitted much differently than a tick-borne virus.  The paper would benefit from focusing on vector-borne diseases in the first paragraph, rather than mentioning viruses that have much different ecological and transmission dynamics than vector-borne diseases.  This seems out of place, and the paper would benefit from the authors focusing on vector borne viruses in the introduction. 

Lines 52-56: Please use proper capitalization for the names of the viruses in this paragraph and throughout the manuscript. If the virus is named after a place/person, the place/person is capitalized (e.g. African swine fever virus), otherwise the name should be lowercase (e.g. tick-borne encephalitis virus).  Please fix throughout the manuscript (including figures and supplementary material).

Figure 1.  This is an interesting depiction of the tick-borne viruses.  I’m assuming the size of each section represents a percentage or at least indicates how many are in each group relative to the rest?  If so, please indicate this in the Figure 1 description.

Line 126.  Assuming “AD” means “Anno Domini”, this should be spelled out the first time used in the manuscript.  However, if there were no viruses mentioned in the manuscript that were “BC”, this seems redundant and can be removed throughout the manuscript for simplicity.

Lines 128-130:  “because they caused human or animal diseases and were discovered after the diseases were detected” – could you please clarify this portion of the sentence.  Do you mean that the viruses detected before 1950 were discovered only because animals/humans were getting ill, and not due to funding?  Could the authors please reword this for clarity?

Figure 2.  Again, this is a very interesting figure, but it does need clarification.  For the viruses abbreviated in Fig. 2A, is this indicating the year the virus was first discovered?  Why are only these 12 viruses listed?  Were there other viruses described later on not listed on the timeline?  Please further describe this in the figure caption. 

Also, please fix capitalization in the names of the viruses in your figure caption.

Line 197: Please be consistent with the number of decimals after the percentages throughout the document for consistency.  For this paragraph, one decimal place is fine, but it would read better if you just rounded up the percentages in this paragraph and omitted the decimal places (e.g. 13%, 75%, 53%, etc.).  It would also get your point across more clearly.  Please consider changing this throughout the document.

Line 235: “related to humans” is vague.  Please clarify.  The previous sentence mentions “pathogens or with significant medical potentials”, but this is also vague. Many of these viruses likely infect more than one organism (which I see is depicted in Fig. 4), and potentially cause illness in more than one organism as well.  Please be more specific with how the authors went about grouping these viruses based on pathogenicity and “medical potentials”.  Perhaps simply stating, “cause infection in humans/rodent/etc.” will clarify this here, as well as in Figure 4.

Figure 4: Please change wording from “related to humans” to something more specific (see above).

Author Response

This manuscript is valuable and well written. The authors present a comprehensive review on tick-borne viruses (TBV), with particular attention on the TBV taxonomy and their association with animal diseases.  The figures were particularly interesting and provide a nice overview of the current standing of TBVs from pathogenic and taxonomic perspectives.  This reviewer has no major comments.  Minor comments/suggestions are listed below.

Minor comments:  

Comments 1. Lines 28-48: The introductory paragraph seems tangential to the topic at hand.  Viruses such as Ebola and Middle East respiratory syndrome are transmitted much differently than a tick-borne virus. The paper would benefit from focusing on vector-borne diseases in the first paragraph, rather than mentioning viruses that have much different ecological and transmission dynamics than vector-borne diseases.  This seems out of place, and the paper would benefit from the authors focusing on vector borne viruses in the introduction.

Response 1: Thank you for pointing this out. We agree with this comment. We have revised “Ebola virus, Middle East respiratory syndrome coronavirus, and severe acute respiratory syndrome coronavirus 2” as “dengue virus, yellow fever virus, and Crimean-Congo hemorrhagic fever virus” in the revised manuscript (Lines 32-33).

Comments 2.Lines 52-56: Please use proper capitalization for the names of the viruses in this paragraph and throughout the manuscript. If the virus is named after a place/person, the place/person is capitalized (e.g. African swine fever virus), otherwise the name should be lowercase (e.g. tick-borne encephalitis virus).  Please fix throughout the manuscript (including figures and supplementary material).

Response 2: Thank you for pointing this out. We have diligently reviewed and revised the virus names in the manuscript, adhering to your guidance. Henceforth, all virus names derived from geographical locations or individual names have been consistently capitalized, while all other virus names have been written in lowercase. This modification has been applied throughout the entire manuscript, encompassing all figures and supplementary materials.

Comments 3.Figure 1. This is an interesting depiction of the tick-borne viruses. I’m assuming the size of each section represents a percentage or at least indicates how many are in each group relative to the rest?  If so, please indicate this in the Figure 1 description.

Response 3: Thank you for pointing this out. We agree with this comment. The size of each section is proportional to the percentage of species for TBVs at the order, family, and genus levels. Specifically, the species section represents one particular species of TBVs. We have indicated the relevant description in the legends of Figure 1 (Lines 103-106).

Comments 4.Line 126.  Assuming “AD” means “Anno Domini”, this should be spelled out the first time used in the manuscript.  However, if there were no viruses mentioned in the manuscript that were “BC”, this seems redundant and can be removed throughout the manuscript for simplicity.

Response 4: Thank you for pointing this out. We agree with this comment. We have removed description of “AD” throughout the manuscript for simplicity.

Comments 5.Lines 128-130: “because they caused human or animal diseases and were discovered after the diseases were detected” – could you please clarify this portion of the sentence. Do you mean that the viruses detected before 1950 were discovered only because animals/humans were getting ill, and not due to funding?  Could the authors please reword this for clarity?

Response 5: Thank you for pointing this out. We agree with this comment. We have revised “because they caused human or animal diseases and were discovered after the diseases were detected” as “which were discovered only because animals/humans were getting ill, and not active discovery of TBVs supported by funding” in the revised manuscript (Lines 135-137).

Comments 6.Figure 2. Again, this is a very interesting figure, but it does need clarification.  For the viruses abbreviated in Fig. 2A, is this indicating the year the virus was first discovered? Why are only these 12 viruses listed? Were there other viruses described later on not listed on the timeline?  Please further describe this in the figure caption. Also, please fix capitalization in the names of the viruses in your figure caption.

Response 6: Thank you for pointing this out. Yes, the abbreviation in Figure 2A represents the year when the virus was first discovered. In the paragraph titled 'Over a hundred year history of discovering TBVs', we focused on describing 12 types of TBVs, hence emphasizing these viruses in Figure 2A. The viruses not mentioned in the paragraph are not shown in Figure 2A, and their first discovery years are described in detail in Supplementary Table 1. To streamline the information, we have included abbreviations for virus names in the paragraph and removed the corresponding descriptions from the legend of Figure 2A (Lines 133-145).

Comments 7.Line 197: Please be consistent with the number of decimals after the percentages throughout the document for consistency. For this paragraph, one decimal place is fine, but it would read better if you just rounded up the percentages in this paragraph and omitted the decimal places (e.g. 13%, 75%, 53%, etc.).  It would also get your point across more clearly.  Please consider changing this throughout the document.

Response 7: Thank you for pointing this out. We agree with this comment. For the consistency of percentages throughout the document, we have rounded up the percentages and omitted the decimal places in the revised manuscript (Lines 107, 156, 202, 205, 210, 236, 360, 361, 395).

Comments 8.Line 235: “related to humans” is vague.  Please clarify.  The previous sentence mentions “pathogens or with significant medical potentials”, but this is also vague. Many of these viruses likely infect more than one organism (which I see is depicted in Fig. 4), and potentially cause illness in more than one organism as well.  Please be more specific with how the authors went about grouping these viruses based on pathogenicity and “medical potentials”.  Perhaps simply stating, “cause infection in humans/rodent/etc.” will clarify this here, as well as in Figure 4. Figure 4: Please change wording from “related to humans” to something more specific (see above).

Response 8: Thank you for valuable suggestions. We agree with this comment. We have removed the vague description (Line 238, and legend of Figure 4) , and revised “related to humans” as “can cause infection in humans” in the revised manuscript (Lines 240-241).